# Reconstructing Synergy-Based Hand Grasp Kinematics from Electroencephalographic Signals

**DOI:** 10.3390/s22145349

**Published:** 2022-07-18

**Authors:** Dingyi Pei, Parthan Olikkal, Tülay Adali, Ramana Vinjamuri

**Affiliations:** Department of Computer Science and Electrical Engineering, University of Maryland Baltimore County, Baltimore, MD 21250, USA; dpei1@umbc.edu (D.P.); polikka1@umbc.edu (P.O.); adali@umbc.edu (T.A.)

**Keywords:** hand kinematics, EEG, neural decoding, hand synergies, brain-machine interfaces

## Abstract

Brain-machine interfaces (BMIs) have become increasingly popular in restoring the lost motor function in individuals with disabilities. Several research studies suggest that the CNS may employ synergies or movement primitives to reduce the complexity of control rather than controlling each DoF independently, and the synergies can be used as an optimal control mechanism by the CNS in simplifying and achieving complex movements. Our group has previously demonstrated neural decoding of synergy-based hand movements and used synergies effectively in driving hand exoskeletons. In this study, ten healthy right-handed participants were asked to perform six types of hand grasps representative of the activities of daily living while their neural activities were recorded using electroencephalography (EEG). From half of the participants, hand kinematic synergies were derived, and a neural decoder was developed, based on the correlation between hand synergies and corresponding cortical activity, using multivariate linear regression. Using the synergies and the neural decoder derived from the first half of the participants and only cortical activities from the remaining half of the participants, their hand kinematics were reconstructed with an average accuracy above 70%. Potential applications of synergy-based BMIs for controlling assistive devices in individuals with upper limb motor deficits, implications of the results in individuals with stroke and the limitations of the study were discussed.

## 1. Introduction

Globally, stroke and spinal cord injury are the most common causes of upper limb paralysis. For many of these individuals, the ability to perform simple activities of daily living (ADLs) may be permanently lost. With extensive rehabilitation, gross arm movements are commonly regained. However, hand paralysis is less responsive to therapy, often resulting in permanent disability. As high as 60% of individuals with paralysis return home without functional use of their paretic hand [1]. Limitations in intrinsic recovery from paralysis necessitate the development of neuroprosthetics, wearable exoskeletons and orthotic devices, and the means to control them to attain optimal performance in prehension. Researchers including our group have studied synergy-based brain-machine interfaces (BMIs) and used them in control of hand exoskeletons as a means to offer assistive devices and rehabilitation [2,3,4]. All these studies are based on a hypothesis presented by Nikolai Bernstein [5], where the central nervous system (CNS) simplifies the control and coordination of the musculoskeletal system with high degrees of freedom (DoFs) in a low dimensional space using synergies.

Research in human sensorimotor control supports the idea that the CNS simplifies the control of high-dimensional DoFs and thereby reduces the complexity of motor control by using synergies [6,7,8]. The experimental evidence from human studies suggests that the hand grasp synergies can be decoded from invasive and noninvasive neural recordings [9,10,11]. Transcranial magnetic stimulation (TMS) research has revealed that rather than the development of new patterns, the evoked finger movements were reorganized with pre-existing primitive patterns represented in the corticospinal system [12]. The study on current source (CS) signals derived from electroencephalography (EEG) has shown that the primary motor cortex is the primary area for controlling finger movement, and characteristic differences between various finger movements were reflected in CS synergies patterns [13]. It has been demonstrated that synergies can offer insights in improving motor control and motor rehabilitation [14,15]. Therefore, developing models of synergy-based hand movements and introducing them in BMIs can be significantly useful in machine-assisted motor learning and movement rehabilitation. To allow movement flexibility and to reduce the high complexity of motor control, especially neural control, synergies have been used to provide an optimal control strategy in controlling hand prosthetics and exoskeletons [2,16].

Grasping is one of the fundamental methods of interacting with the environment around us. Non-invasive EEG provides efficient and reliable applications in many fields when paired with advanced signal processing algorithms [17], including EEG-based BMI systems. Over the past years, EEG-based BMIs were able to offer control of external devices and have played a critical role in motor control and motor rehabilitation in individuals with spinal cord injury [18,19] and stroke [20,21,22,23,24]. Most of them are machine-assisted control and learning of hand movements of activities of daily living by decoding human intent from scalp EEG signals. The decoding of EEG correlates of movement intentions has shown the efficiency of BMI-assisted stroke rehabilitation [25]. Agashe et al. decoded reach-to-grasp decoding using low-frequency of EEG [10] and successfully realized robotic hand control by amputees [26]. However, these BMIs based on multiclass intention detection rely on the imagination of basic motor tasks. Thus, researchers started to decode the movement trajectories to realize a more flexible hand grasp by reconstructing the movement trajectory of upper limb movement using time-series EEG [27,28]. It has been confirmed that it is possible to obtain both movement directions [29] as well as kinematic information from low-frequency EEG signals [30]. Mondini et al. successfully transferred offline trajectory reconstruction to continuous online robotic control [31], providing the probability of real-time natural control directly using non-invasive brain activity. Although these studies are encouraging, the high complexity of the human hands has challenged offline and online movement decoding.

One of the current challenges of BMIs is modeling the complex relationships between high-dimensional brain activity, high-dimensional movement kinematics and the underlying movement primitives. Such models can enable dexterous motor control to help restore near natural hand movements for those who lost their hand functions, and also offer new methods of neuromotor rehabilitation. In a related previous study by our group, the hypothesis that the CNS may modulate the combinations of synergies, to realize synergy-based motor control was tested and the results successfully captured the correlations between neural features and the hand kinematic synergies for each individual [32]. However, for individuals with paralysis and other movement disorders who cannot perform physical movements, it is challenging to establish such a correlation model. Therefore, this paper focused on the study of generalization of the correlation between the combination of hand kinematic synergies and corresponding neural activity from the training subject group, and reconstructed the hand kinematics for the testing subject group using only their neural activity based on the correlation model. Potential applications of neural decoding based on the synergies are expected to assist efficient BMIs in machine-assisted motor control and rehabilitation.

The remainder of this paper is organized as follows. Section 2 gives the detailed elaboration of experimental design (Section 2.1 and Section 2.2), and methodologies of synergy derivation and neural feature extraction (Section 2.3, Section 2.4 and Section 2.5). Section 3 presents the results and analysis and Section 4 discusses the implications of the study, potential applications, limitations, challenges and leaves the reader with possible future alternatives and solutions.

## 2. Materials and Methods

This section described experimental design and the detailed methods used to achieve the research objectives. Section 2.1 and Section 2.2 described experimental protocol and data collection. Section 2.3, Section 2.4 and Section 2.5 described computational methods behind the derivation of kinematic synergies, extraction of neural features and computation of cortical correlates of kinematic synergies.

### 2.1. Experimental Protocol

Ten healthy, right-handed individuals (4 males, 6 females; mean age 23.0 ± 3.1 years) participated in this study and they were asked to perform the six types of hand grasp movements representative of their activities of daily living as shown in Figure 1A. There were six objects selected to represent different grasps, including a screwdriver (tripod grasp), water bottle (cylindrical grasp), CD (lateral grasp), petri dish (spherical grasp), handle (hook grasp) and bracelet (precision grasp). At the beginning of the experiment, the subject sat and relaxed in front of the experimental table with the dominant hand flat with their palm facing down on the initial place on the table as indicated. The start and stop of the movement were indicated by auditory stimuli. Their hand movements and corresponding EEG were recorded simultaneously. At the ‘start’ beep, the subjects rapidly grasped and held the object until a ‘stop’ beep. The task timeline is provided in Figure 1B. Each object was presented in random order and each grasp was repeated 30 times, for a total of 180 grasping tasks per subject. During the recording, subjects were asked to refrain from blinking or swallowing to restrict the artifacts as far as possible. The trials with eye movements/blinks and muscle artifacts were rejected.

### 2.2. Data Collection

Data was collected under an approved protocol by the Internal Review Board (IRB) of the Stevens Institute of Technology. During the experiment, the hand kinematics were recorded using a CyberGlove (CyberGlove Systems LLC, San Jose, CA, USA) worn on the subjects’ dominant hand. In this study, 10 of 18 sensors were used to measure the metacarpophalangeal (MCP) joints of the thumb and four fingers, interphalangeal (IP) joint for the thumb and proximal interphalangeal (PIP) joints of the four fingers. Each subject also performed initial postures (flat and fist) to calibrate the sensors of CyberGlove. CyberGlove data was captured at a 125 Hz sampling rate using a custom-built LabVIEW (National Instruments Corporation, Austin, TX, USA) program. This custom-built LabVIEW program also provided the audio cues and sent a synchronizing wave to the EEG recording amplifiers to align hand kinematics and EEG. For neural signal acquisition, EEG was recorded by a high-density EEG cap with 32 active electrodes based on 10/20 system positions with an additional 86 intermediate positions (g.GAMMA cap, g.tec, Schiedlberg, Austria). These electrode locations are F3, Fz, F4, FC3, FC1, FC_Z_, FC2, FC4, C5, C3, C1, C_Z_, C2, C4, C6, CP3, CP1, CP_Z_, CP2, CP4, P3, P_Z_, P4, O_Z_ covering frontal (F), central (C), parietal (P) and occipital (O) areas. Eight other intermediate locations were also recorded as shown in Figure 1C. A ground electrode was placed at the nasion (yellow) and the reference (Ag, AgCl) electrode was on the right or left ear lobe. A conductive gel (g.GAMMAgel) was used to place each electrode (g.Ladybird). Impedance was kept below 5 kOhms and checked throughout the experiment. Data were continuously captured with two amplifiers (g.USBamps) using BCI2000 [34] with a sampling rate of 256 Hz.

### 2.3. Derivation of Hand Kinematic Synergies

Ten subjects were randomly divided into a training set (five subjects) and a testing set (five subjects). The kinematic synergies were derived from movements recorded from the training set. The hand kinematics (joint angular velocities) from 10 joints were derived from the differential of the recorded joint angles, and kinematic synergies were extracted from all types of hand grasps according to the synergy-based hand movement model discussed in [35] and as discussed here briefly. The hand kinematics $v\left(t\right)$ can be expressed as a linear combination of kinematic *n* synergies $S\left(t\right)$ and corresponding synergy weights *c* (1), where the synergies contain both spatial and temporal characteristics of the hand movement. The movement can also be represented as a matrix and be decomposed using singular value decomposition (SVD, as shown in (2))
(1)$$v\left(t\right)=\sum_{j=1}^{n}c_{j}S^{j}\left(t\right)$$
(2)V=UΣS=CS
where ***V*** is the matrix of angular velocities with dimensions *m* × *n*, where *m* is the number of hand movements in training set and *n* is the concatenated angular velocities of 10 joints. ***S*** contains the eigenvectors, or principal components (PCs), which are considered as synergies. ***C*** is a matrix that contains weights assigned to synergy ***S***. A total number of six synergies were selected, as provided in Figure 2, depending on the variance at an 85% of threshold. The fraction of the variance of the synergies and reconstruction performance with the selected number of synergies is illustrated in Figure 3.

### 2.4. Extraction of Neural Features

The baseline of the recorded raw EEG data was corrected by subtracting the mean value of resting EEG signals, and the linear trend of EEG was removed. EEG signals were then filtered to 0.1–56 Hz frequency range. Figure 4 shows the EEG modulation after the audio stimulus. Post-stimulus desynchronization occurred first, followed by post-stimulus synchronization. Neural activity gradually settles down accompanied by the end of movement two seconds post-stimulus. Thus, the dynamic changes from the two-second post-stimulus period were captured by a 500 ms wide sliding window with a 375 ms overlap. The EEG features were extracted by computing the average power of the integral of the power spectral density (PSD) estimate within each sliding window. The 500 ms width was used to obtain a higher spatial resolution in the frequency domain. Since only two seconds were selected from stimulus onset to end of the movement, the 75% overlap was to ensure as much information as possible was included. A total of 13 EEG features were extracted from each electrode. Since 32 EEG electrodes were recording the neural activity simultaneously, the principal component analysis (PCA) was applied to extract top-ranked components as neural features which shared common features across multiple electrodes.

### 2.5. Cortical Correlates of Kinematic Synergies

Although the synergies vary across different individuals depending on their own unique movement primitives, previous studies have shown [36,37] that spatiotemporal patterns of synergies are not only shared across a different set of motor behaviours but also carry generalized control strategies across individuals. Hence, in this study, kinematic synergies were extracted from half of the subjects, and used for the other half of the subjects, for neural decoding and synergy-based reconstruction of their movements. The decoding paradigm is shown in Figure 5. The brain and behaviour data collected from ten subjects were randomly divided into two halves with five subjects in each called the training set and the testing set. The training set was used to derive the kinematic synergies and model the linear correlation matrix between the synergy weights and corresponded neural features. In the testing phase, by using the extracted kinematic synergies and by applying the trained coefficient matrix from the training set, the synergy weights for the testing group were decoded from their neural features and their hand kinematics were reconstructed. The performance was estimated from the correlations between the recorded kinematics and decoded kinematics.

Previous studies [38,39,40] applied linear regression for modeling the hand movements, where each joint movement is independently regarded as a linear combination of neural activities in the time domain. Studies from our lab [32,34] have successfully correlated the neural features with the coefficients of the synergies. In this paper, the neural decoding model was realized using multivariate linear regression to determine the relationship between neural features and the weights of synergies from the training set:(3)C=Xβ
where ***C*** is a matrix the weights of synergies, and for each task, ***C*** is an *n* × *m* matrix, where *n* is the number of movement trials and *m* is the number of synergy weights. ***X*** represents an *n* × *p* neural feature matrix, and *p* denotes the number of EEG features. ***β*** is a *p* × *m* coefficient matrix. The weights of synergies from the testing set were calculated using corresponding neural features through this multivariate linear regression model, and the hand kinematics were then reconstructed by linearly combining the decoded synergy weights and the kinematic synergies extracted from the training set. Decoding accuracy between the recorded kinematics and neural decoded kinematics was measured using the Pearson correlation coefficient (ρ). The decoding error was defined as 1 − |ρ|. To avoid the bias from certain group of subjects, this model was evaluated with 10-fold cross-validation with shuffled subjects in the training set and testing set each fold.

In order to investigate how each synergy was modulated by a specific cortical area, the independence of the EEG electrodes used for the decoding process was evaluated. The neural features were extracted using the same procedure from each single EEG electrode and the nearest electrodes, followed by the linear projection to corresponding synergy weights. The projection matrix includes the correlation coefficients of each set of neural features with the weights of each synergy. The larger magnitude represents a higher correlation between the synergy weights and neural activity. Thus, the independence of each area was interpreted by the following model:(4)$$D_{n}=\sum_{k=1}^{K}\sqrt{\sum_{m=1}^{M}\beta_{mk}^{2}}$$
where *D*_n_ represents neural independency density of the *n*th synergy, *m* is the number of regression parameters ***β*** calculated above and *k* is the number of hand grasp tasks. The variables and acronyms in Section 2 are defined in Table 1.

## 3. Results

The kinematic synergies and neural decoding model were obtained from the training group of subjects, the weights of the synergies from the testing group of subjects were calculated and their hand kinematics were decoded. Since subjects were randomly grouped, the results were verified with cross validation and the hand kinematics were reconstructed from neural representations with an average decoding accuracy of 70.8 ± 5.5% across all movement tasks and all individuals. Table 2 shows the results of the decoding accuracy represented by the correlations between the recorded kinematics and neural decoded kinematics. This illustrated the performance of each movement task and the accuracy of each individual from one of 10-fold cross-validation. Figure 6 provides the comparison of recorded and decoded kinematics patterns of ten joints (from Subject 2, spherical grasp). Results show the averaged trajectories across 30 repetitions (solid lines), where the deviation of different repetitions varies (shaded areas). The decoding model was able to tune weights differently for different movements and thereby successfully decode the diverse angular velocity patterns across different objects. The reconstruction patterns for six types of grasps are illustrated in Figure 7. The corresponding kinematic trajectories of key digits involved in all types of grasps (thumb, index and middle fingers) were predicted accurately especially in lateral grasp (object: compact disc) and precision grasps (object: bracelet).

Figure 8 provides averaged neural-based decoding errors for each joint and grasp type. Thumb, index and middle are dominant fingers in a variety of grasps, but the extent of IP/PIP joint varies depending on individual grasp natures. Thus, it can be observed in Figure 8A that the MCP joints of the thumb, index and middle are lower in error than other joints, whereas the IP/PIP joints of these three fingers are relatively poor in decoding. Surprisingly, PIP joints of the ring and pinky fingers have relatively lower errors than MCP joints, which may be due to the fact that these two fingers have more freedom in grasping. Figure 8B provides the decoding performance of six grasp types. The lowest decoding error was observed from the lateral grasp, and the tripod grasp (object: screwdriver) and spherical grasp (object: petri dish) have relatively higher errors among all grasp types.

How each synergy potentially correlates to a certain neural activation area is shown in Figure 9. Results were averaged across movement tasks and individuals from the testing set after 10-fold cross-validation. It is observed that each synergy has different topographical representations and shows diverse electrode activations. Since all subjects were right-handed, the hand synergies were highly modulated on the central and left hemisphere, covering the areas of movement intention and motion-related activation generation. Specifically, Synergies 1 to 5 were effective spatially with respect to hand area, and span across the premotor cortex, primary motor cortex and primary sensorimotor cortex. Synergies 1, 4 and 6 were largely activated in the primary motor areas and supplementary motor areas that control the dynamics of movement [41]. When performing simple movements, the primary motor cortex and the primary somatosensory cortex were activated on the contralateral hemisphere, as observed from Synergies 1, 3 and 4.

## 4. Discussion

The central hypothesis of this paper was that the hand kinematic synergies and a neural decoder derived from a group of healthy individuals with no movement disorders can be generalized and used in reconstructing high dimensional hand movement kinematics solely based on neural activities in individuals who cannot perform hand movements. In this paper, the study participants were first divided into two groups—training and testing. From the training group, hand kinematic synergies were derived and a neural decoder was developed. Using the synergies and neural decoder from the training group and only neural activities from the testing group, their hand kinematics were reconstructed. This supports potential applications of non-invasive synergy-based brain-machine interfaces in neuromotor control and rehabilitation of individuals with upper limb motor deficits due to stroke or spinal cord injury.

### 4.1. Movement Decoding from EEG

Noninvasive EEG-based BMIs have been developed to decode user’s movement intention based on the markers of active brain involvement in the preparation of the desired movement. Previously, studies have decoded hand kinematics using a linear regression model for individual joints from low-frequency EEG signals (<1 Hz) [38]. Mondini et al. [31] used low-frequency EEG signals to demonstrate offline as well as online continuous closed-loop control of a robotic arm. Global cortical EEG activity was used to predict the shape of hand during grasping [10]. Other studies have also showed that the best performance of motor imagery was identified from mu and beta frequency bands of EEG [28]. Nonlinear models were also used to achieve a robust decoding model for finger movement decoding [13]. These studies strongly support that the neural correlates of fine finger movements can be extracted from EEG signals.

### 4.2. Neural Representations of Synergies

Initial attempts to detect the neural bases of grasp synergies were conducted in non-human primates [42,43,44,45,46] and humans [6,8,9,47,48]. In direct relevance to this study, Gentner and Classen [8] observed structured similarities between postural synergies derived from the TMS-evoked finger movements and the postural synergies derived from voluntary movements. In a recent study [10], researchers demonstrated that they can achieve up to 60% decoding accuracies in predicting three hand grasping kinematic synergies from scalp EEG signals. In [11,49], neural responses during grasping were measured by functional magnetic resonance imaging (fMRI). The results favored kinematic synergy-based control encoded in motor cortical areas than muscle synergy-based control. Although the question is open whether muscle-synergy models or kinematic-synergy models are encoded in CNS [50], nevertheless, all studies seem to favor the presence of synergies in CNS.

In this study, the synergy-based hand kinematics were successfully decoded with an average accuracy of 70% from EEG spectral features using a multivariate linear regression model. Our model is based on the correlation between the neural features and the weights of kinematic synergies. Our hypothesis emphasizes that the synergies are present in the lower level neural and musculoskeletal systems and the weights of synergies are encoded in the higher-level neural systems. Thus, for achieving any movement, the higher-level neural systems can tweak the weights of the synergies instead of controlling individual joints. The CNS may regulate different combinations of synergies to achieve various hand grasps. The neural features used in this study were from a broad frequency range of EEG signals (0.1–56 Hz). This encompasses compensations from other bands for abnormal mu/beta waves which might be the case in individuals with stroke or other neurodegenerative diseases. Furthermore, the synergies from able-bodied individuals were generalized across individuals among the second group whose kinematics were not considered. This methodology can help in the neural decoding of hand movements in individuals with motor disabilities.

### 4.3. Relevance to Individuals with Stroke

As BMIs create a strong link between the brain (EEG in this study) and behavior (kinematic synergies in this study), they can be significantly helpful in motor control as well as motor rehabilitation. Thus far, several BMIs have been proposed for neural decoding of movements in individuals with stroke. However, high-dimensional movement decoding using EEG is still an open challenge. To our best knowledge, this is one of the few studies that address high-dimensional movement decoding using low-dimensional and low-resolution non-invasive EEG signals by introducing a synergy-based movement generation model. Furthermore, the performance of EEG-based movement decoders is dependent on the type of decoding algorithms used. In this study, the synergy-based algorithms were restricted to simple linear decoders that can perform well in real-time as well.

In this study, all participants were healthy. Thus, no lesions or abnormalities were present in the recorded EEG that was used for decoding the weights of synergies. Question arises whether these results are applicable in individuals with stroke. Functional reorganization of the motor system after stroke has been documented [51]. It has been shown that stroke lesion location influences the performance of the decoder [52]. Abnormal changes of mu and beta waves were observed in stroke patients [22,23,24], and premovement activity <5 Hz was diminished after acute motor stroke [53]. However, in individuals with stroke, bilateral EEG has been successfully used for movement decoding. After stroke, patients have bilateral or enhanced contralesional cortical activity during motor tasks [54]. It was also demonstrated that patients shifting brain activation towards ipsilesional areas experienced motor recovery [25]. The changes in the alpha/beta rhythms of EEG successfully assisted BMIs to detect the motor intentions in stroke patients [25] and decode upper limb movements [55]. All these studies hold promise that synergy-based movement can be decoded from scalp EEG. In the near future, the current study will be repeated on individuals with stroke.

The suppression of sensorimotor rhythms in individuals with stroke (in the range of mu 8–12 Hz and beta 18–30 Hz bands) before and during the movement could possibly affect the decoding accuracy of EEG-based BCIs. How the synergy modulation is influenced by these changes in rhythms needs further investigation. Based on recordings from healthy subjects in this study, Figure 10 illustrates neuromodulation for each synergy at different EEG rhythms. In the primary motor areas, the synergy-based decoding was strongly affected by mu and beta bands, and in the prefrontal, somatosensory and supplementary areas, the decoding was strongly affected by theta and gamma bands. Thus, considering bihemispheric broad band EEG activity could compensate for suppressed neural signals due to lesions.

### 4.4. Limitations and Future Research

This study demonstrated that the hand synergies (principal components that captured highest variance) and neural decoder (a multivariate linear regression model) derived from a group of healthy individuals with no movement disorders can be generalized and used in modeling high hand dimensional movements in a new group of individuals solely based on their neural activities. Only ten healthy, right-handed participants were included in the experiment, relatively limiting the diversity and stability of the linear model among a large group of individuals. In the near future, the study will be extended to a larger group of participants and will be repeated in individuals with movement disorders such as paralysis due to stroke or spinal cord injury. Additionally, most of the BMIs control assistive devices by the neural activity derived from imagined movement rather than executed movement, especially for individuals with paralysis who cannot execute a movement. Thus, the practicability and generalizability of hand kinematic reconstruction and generation based on the proposed model will be verified in the future research.

## 5. Conclusions

In this study, synergy-based hand movements were reconstructed using neural correlates of hand synergies found in spectral features extracted from broad band EEG activity. Although the exact cortical locations of neural representations of synergies could not be pinpointed, which could be due to the poor spatial resolution of scalp EEG, the results nevertheless achieved 70% accuracy in neural decoding of synergy-based movements. By using the kinematic synergies and a linear neural decoder developed within a group of training subjects, the hand grasping movements in a group of testing subjects were successfully decoded solely based on their neural activities. To our best knowledge, this is one of the few studies that attempted to decode high-dimensional hand movements using low-dimensional and low-resolution EEG activities. Based on the neural representations of synergies obtained from multiple frequency bands and multiple cortical areas, it appears that considering bihemispheric EEG activity combined with synergy-based movement decoding can yield functional control of synergy-based prostheses and exoskeletons. Thus, future work will focus on the verification of whether the results presented in this study will transfer to individuals with stroke. The neural decoding model presented in this paper will also be validated in a real-time synergy-based control.

## Figures and Tables

**Figure 1 sensors-22-05349-f001:**
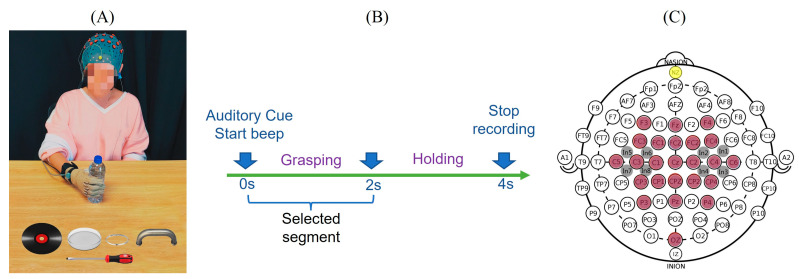
Experimental setup, electrode placement and the protocol. (**A**) The subject was asked to perform six types of hand movements by grasping six different objects including a water bottle, a CD, a petri dish, a screwdriver, a bracelet and a door handle. During the movement, both hand kinematics and EEG activities were recorded. (**B**) The start of hand movement was instructed by an auditory cue, and the subjects grasped and held the object until the ‘stop’ cue. (**C**) Neural signals were recorded using 32 electrodes including 24 denominative electrodes (in brown) and 8 intermediate locations (in grey). Adapted from [32,33].

**Figure 2 sensors-22-05349-f002:**
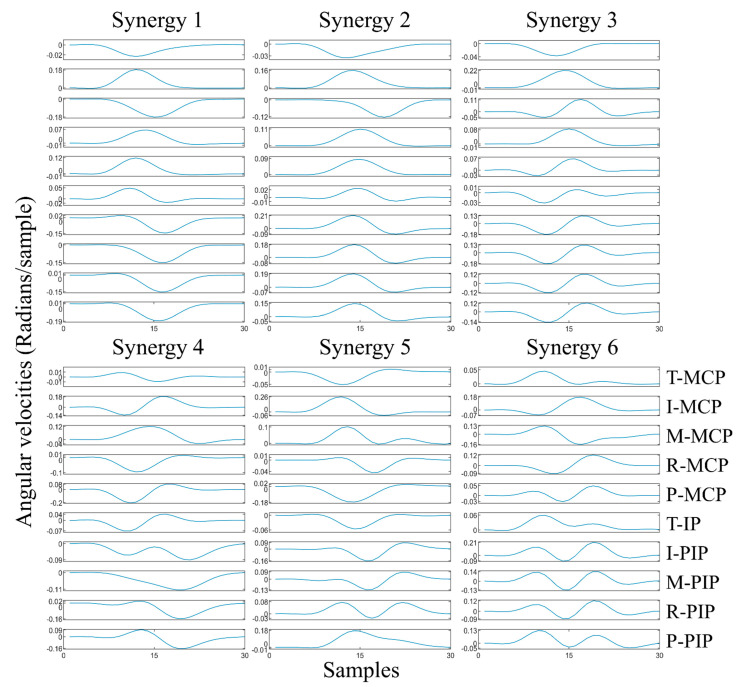
Six kinematic synergies of each hand joint derived from the training group of 5 subjects. T: thumb, I: index, M: middle, R: ring, P: pinky, MCP: metacarpophalangeal, IP: interphalangeal, PIP: proximal interphalangeal.

**Figure 3 sensors-22-05349-f003:**
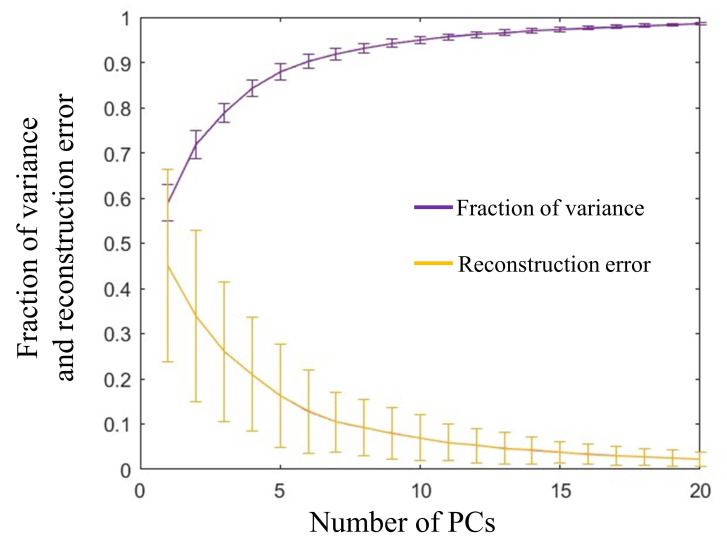
Fraction of variance of PCs and reconstruction error using different numbers of PCs.

**Figure 4 sensors-22-05349-f004:**
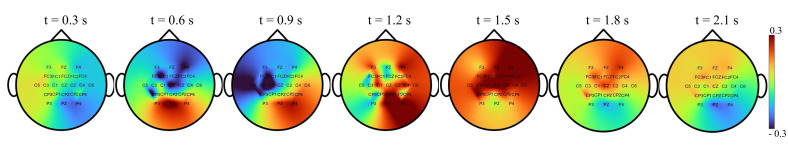
Dynamic neural activity changes during grasp. The averaged EEG activity amplitude from motor areas in all movement tasks. Post stimulus (t = 0 s) activity in the motor cortex first decreases after post-stimulus and then increases to peak during the movement (the color bar indicates the normalized amplitude of EEG activity).

**Figure 5 sensors-22-05349-f005:**
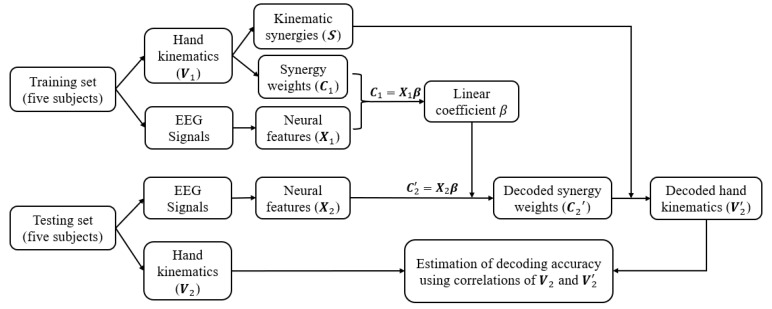
The decoding methodology. Ten subjects were randomly divided into two equal groups for training and testing. The training set of five subjects was used to derive the kinematic synergies and model the linear correlation between the synergy weights and corresponding neural features. In the testing phase, by using the coefficients and extracted kinematic synergies from training set, the synergy weights were decoded from neural features and the hand kinematics were reconstructed for testing set. The performance of the neural decoded and reconstructed kinematics was estimated from the correlations between the recorded kinematics and reconstructed kinematics.

**Figure 6 sensors-22-05349-f006:**
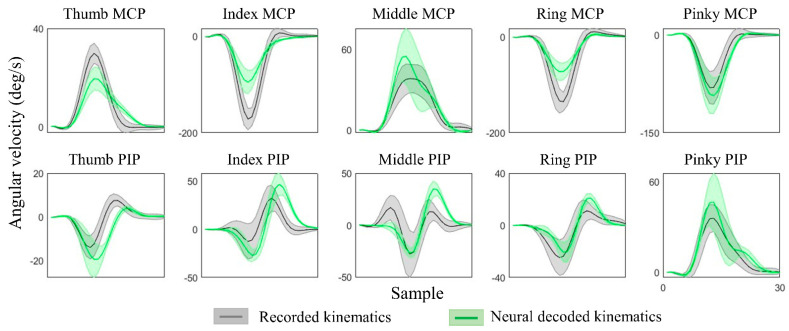
Recorded and neural decoded joint angular velocities of PIP and MCP joints of thumb and four fingers. The recorded kinematics (in grey) and neural decoded kinematics (in green) are averaged across all repetitions for Subject 2, Task 4, and the shaded regions represent standard deviations.

**Figure 7 sensors-22-05349-f007:**
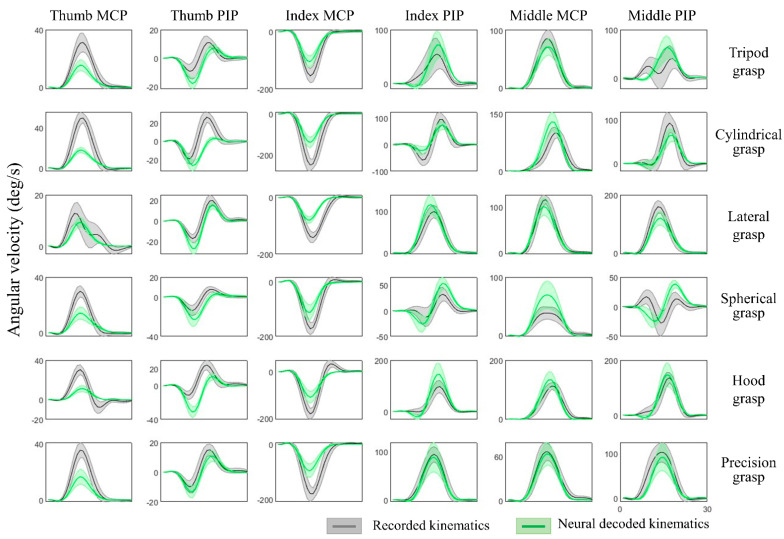
Hand joint angular velocities of six types of grasps and joints from thumb, index and middle. The recorded kinematics (in grey) and neural decoded kinematics (in green) are averaged across all repetitions from Subject 2, and the shaded regions represent standard deviations. The ring and pinky finger are not shown here for simplicity in illustration.

**Figure 8 sensors-22-05349-f008:**
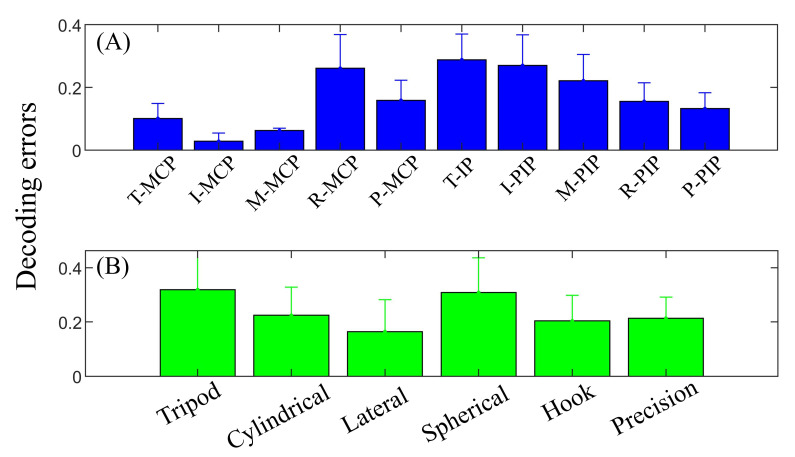
Decoding error based on ten joints (**A**) and six grasp types (**B**). MCP joints of the thumb, index and middle have lower errors than respective IP/PIP joints, whereas MCP joints of the ring and pinky have relatively higher error.

**Figure 9 sensors-22-05349-f009:**
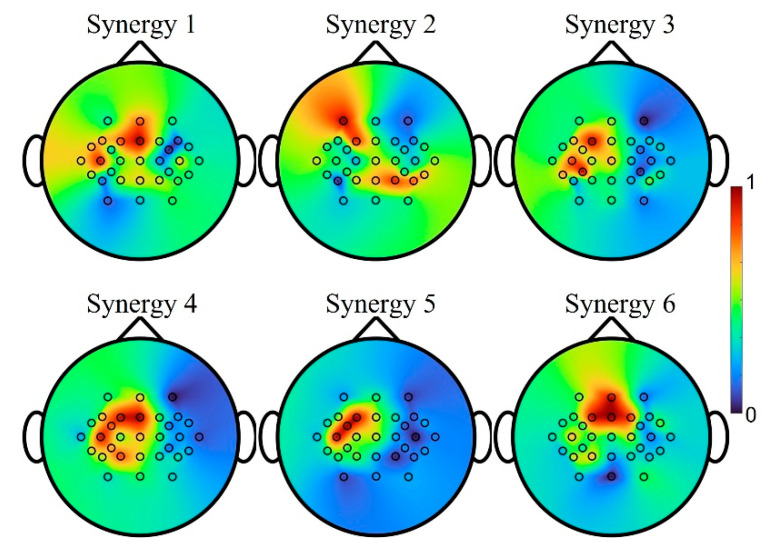
Spatial correlations of each synergy reflected on the scalp tomograph for Subject 2. The hand synergies are highly modulated on the central and left hemisphere, covering the areas of movement intention and motion-related activation generation. Synergies 1 to 5 were effective spatially with respect to hand area, and span across the premotor cortex, primary motor cortex and primary sensorimotor cortex. Synergies 1, 4 and 6 were mainly activated on the primary motor area and supplementary motor area. The primary motor cortex and the primary somatosensory cortex were activated on the contralateral hemisphere, as observed from Synergies 1, 3 and 4.

**Figure 10 sensors-22-05349-f010:**
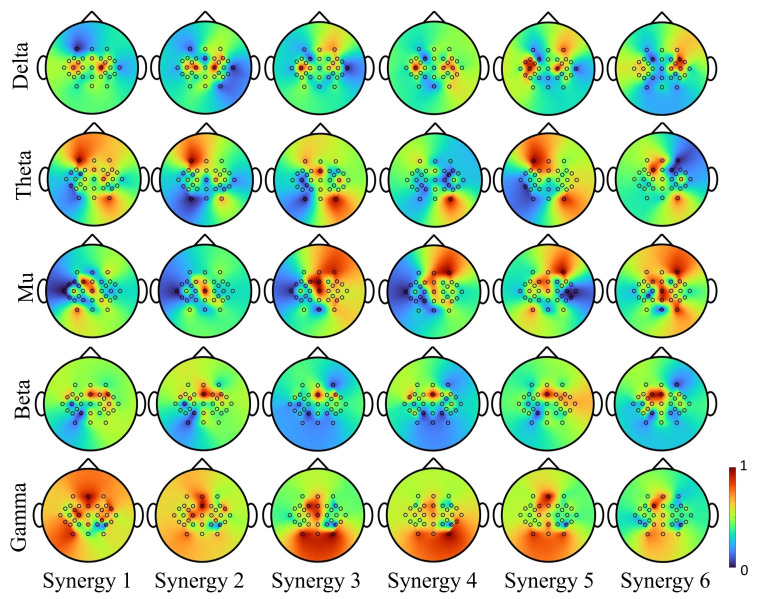
Synergy modulation in delta, theta, mu, beta and gamma frequency bands of EEG. Synergies activation mainly concentrated on the primary motor area modulated in low-frequency rhythm (delta). In theta rhythm, strong activation was observed on the contralateral premotor cortex, and ipsilateral somatosensory cortex. Mu modulations were observed in the supplementary motor cortex, and beta were observed in the bilateral primary motor region. Activations observed from gamma spanned over all motor-related areas.

**Table 1 sensors-22-05349-t001:** Nomenclature table (symbols and definitions).

v	Joint angular velocities
V	Matrix of joint angular velocities
c	Synergy weights
C	Matrix of synergy weights
S	Kinematic synergies
S	Matrix of kinematic synergies
X	Matrix of neural features
β	Regression coefficients
D	Neural independency density of the synergy

**Table 2 sensors-22-05349-t002:** Correlations between the neural decoded kinematics and recorded kinematics (Mean ± Std).

	Subject 1	Subject 2	Subject 3	Subject 4	Subject 5
Task 1	72.0 ± 8.0%	65.1 ± 11.2%	56.7 ± 20.5%	69.2 ± 8.2%	79.4 ± 7.1%
Task 2	84.4 ± 9.3%	74.1 ± 10.0%	77.5 ± 6.2%	77.7 ± 9.3%	74.1 ± 12.8%
Task 3	89.9 ± 6.3%	83.7 ± 14.8%	80.9 ± 13.2%	84.5 ± 10.4%	78.7 ± 9.3%
Task 4	75.5 ± 6.4%	59.3 ± 12.9%	75.1 ± 7.5%	72.2 ± 9.8%	64.6 ± 15.4%
Task 5	84.3 ± 7.0%	82.2 ± 7.2%	82.1 ± 8.7%	70.5 ± 10.3%	79.8 ± 7.1%
Task 6	77.4 ± 6.6%	81.0 ± 6.4%	80.1 ± 7.7%	73.7 ± 7.4%	80.8 ± 8.1%

## Data Availability

Not applicable.

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
