# Peer review of "Reconstructing Synergy-Based Hand Grasp Kinematics from Electroencephalographic Signals"

_sensors, 2022, doi:10.3390/s22145349_

Round 1

Reviewer 1 Report

The authors present the article entitled “Reconstructing Synergy-based Hand Grasp Kinematics from Electroencephalographic Signals”.

This paper focuses on the study of the correlation between the hand kinematic synergies and corresponding neural activity using a multivariate linear regression model. they expect neural decoding based on the synergies will lead to efficient BMIs in machine-assisted motor control and rehabilitation. 

Line 47-50 could be justified by the following EEG works: Impact of eeg parameters detecting dementia diseases: a systematic review;  Comparative study of time and frequency features for eeg classification

The article presents the following concerns:

  • Line 31: Reference is missing.

  • Avoid using first-person sentences. Use third-person or passive voice sentences.

  • Line 36: Please read the guide for authors for grouped citations. Same in line 328.

  • Line 78: Improve the objective of the work. I suggest mentioning the main contributions of the work by highlighting its novelty of the work.

  • Add a little introduction between points 2 and 2.1

  • Figures must be vectorized.

  • Please add future works in the Conclusion section.

  • The final part of the introduction should make a description of the structure of the text.

  • References 1 and 15 are the same.

  • Add hyperlinks to tables, figures, and references.

  • Please add a nomenclature table to define variables and acronyms.

  • Justify the relevance of this article with references from the journal.

  • My biggest concern is that the work presents many coincidences with the next article and isn't referenced:

D. Pei, V. Patel, M. Burns, R. Chandramouli and R. Vinjamuri, "Neural Decoding of Synergy-Based Hand Movements Using Electroencephalography," in IEEE Access, vol. 7, pp. 18155-18163, 2019, doi: 10.1109/ACCESS.2019.2895566.

Reviewer 2 Report

1. The number of subjects in this paper is samller. Is it possible to add more subjects in this study?

2. why EEG signals were first filtered to 0.1-56 Hz frequency range ? Is it any preprocessing for EEG ? In general, there are some nosie in the EEG. 

3. In Figure 4, the channels are not fit well in the head scalp.

4. This study adopted 10-fold cross-validation, but the size of training set and testing set are same. Is it he 2-fold cross-validation ?

Reviewer 3 Report

The quality of paper is good, however, some issues should be addressed

q- the limitations of the approach must be recognised. 

2-The authors should clearly highlight the novel technical contributions in the proposed research.

3- What was the ground truth used to evaluate the proposed model. 

Round 2

Reviewer 1 Report

The manuscript can be accepted